# Bioprinted Schwann and Mesenchymal Stem Cell Co-Cultures for Enhanced Spatial Control of Neurite Outgrowth

**DOI:** 10.3390/gels9030172

**Published:** 2023-02-22

**Authors:** Enateri V. Alakpa, Anton Bahrd, Krister Wiklund, Magnus Andersson, Lev N. Novikov, Christina Ljungberg, Peyman Kelk

**Affiliations:** 1Department of Integrative Medical Biology, Umeå University, SE-901 87 Umeå, Sweden; 2Department of Physics, Umeå University, SE-901 87 Umeå, Sweden; 3Department of Surgical and Perioperative Science, Section of Hand and Plastic Surgery, Umeå University, SE-901 87 Umeå, Sweden

**Keywords:** 3D bioprinting, biosynthetic conduit, dorsal root ganglion, mesenchymal stem cells, nerve regeneration, Schwann cells

## Abstract

Bioprinting nerve conduits supplemented with glial or stem cells is a promising approach to promote axonal regeneration in the injured nervous system. In this study, we examined the effects of different compositions of bioprinted fibrin hydrogels supplemented with Schwann cells and mesenchymal stem cells (MSCs) on cell viability, production of neurotrophic factors, and neurite outgrowth from adult sensory neurons. To reduce cell damage during bioprinting, we analyzed and optimized the shear stress magnitude and exposure time. The results demonstrated that fibrin hydrogel made from 9 mg/mL of fibrinogen and 50IE/mL of thrombin maintained the gel’s highest stability and cell viability. Gene transcription levels for neurotrophic factors were significantly higher in cultures containing Schwann cells. However, the amount of the secreted neurotrophic factors was similar in all co-cultures with the different ratios of Schwann cells and MSCs. By testing various co-culture combinations, we found that the number of Schwann cells can feasibly be reduced by half and still stimulate guided neurite outgrowth in a 3D-printed fibrin matrix. This study demonstrates that bioprinting can be used to develop nerve conduits with optimized cell compositions to guide axonal regeneration.

## 1. Introduction

Traumatic injuries to the peripheral nerves are common and in Europe alone the annual incidence of nerve injuries affecting the hand is 140 per million inhabitants [1]. The injuries often affect young economically active individuals and could result in life-long disability with neuropathic pain and permanent loss of sensory and motor functions [2].

Autologous nerve grafts are the standard procedure for repairing peripheral nerve injuries [3], but they require harvesting a functional nerve, resulting in further neurological deficit [4]. Furthermore, although short-range nerve gaps are amenable to repair using nerve grafts, large nerves and critical-sized defects (3 cm or greater in humans) are more challenging, as the supply of a sufficient amount of nerve grafts may be impossible. In such situations, an alternative approach may be to use biosynthetic nerve conduits. In addition to providing physical constraint or guidance, the conduits should preferably be biodegradable and incorporate biochemical cues such as extracellular matrix (ECM) molecules [5,6,7], growth factors [8,9], and various supporting cell types [5,7,10,11].

Schwann cells are the endogenous glia cells that promote peripheral axon regeneration and, therefore, can be used to facilitate nerve repair [12]. They, however, suffer the caveat of requiring the use of a donor nerve, as well as the need for relatively long expansion times in vitro, during which inherent properties could become altered or lost [8,13]. Adult stem cells/mesenchymal stem cells (MSCs) sourced from several tissue types including fat [14], bone marrow [15], and dental pulp [16] are relatively more accessible than Schwann cells and can be stimulated to exhibit Schwann cell-like properties. Furthermore, in vivo studies indicate that MSCs can augment axonal regeneration [10,15,17,18]; however, they often fail to reach the full potential of Schwann cells. Current research to compensate for this problem has examined the use of co-cultures of alternative cell types that are able to produce sufficient amounts of neurotrophic molecules to support the survival of neurons and axonal growth [19,20].

Bioprinting is a facet of 3D printing that incorporates the use of cells, biomaterials, and/or bioactive molecules to construct scaffolds with high design reproducibility and precision. This ability to bring together the properties of physical constraint (spatial specificity/precision) and biocompatibility makes bioprinting a tool with enormous potential for automating the creation of physically and biologically instructive materials with high throughput, being both time- and cost-effective. In comparison to the traditional protocol of first printing a scaffold prior to cell seeding, bioprinting allows for the simultaneous and precise placement of cells. Hence, it is possible to integrate the cellular component into the scaffold framework and design. Bioprinting of tissue-engineered constructs as neural implants allows an increased potential for promoting regeneration across large nerve gap defects as it is a means of combining scaffold and support cells using the desired specific designs within a singular conduit [21,22].

However, the printing process can affect cell viability and production of neurotrophic factors (NTFs) important for the stimulation of neuronal survival and regeneration [23,24]. As such, bioinks or the matrix used for cell delivery is an important factor, and this must both support cell survival and be of a ‘printable’ consistency. The optimal matrix consistency for 3D printing and for allowing migration of cells through a conduit is typically at odds with one another. Thus, there is a need to develop a matrix that enables 3D printing and allows cell migration [25].

In this study, we optimized and evaluated the bioprinting technique using a diluted fibrin matrix supplemented with cultured Schwann cells and MSCs to enhance directed neurite outgrowth from the adult sensory dorsal root ganglion (DRG) neurons. In addition, we assessed the possibility of reducing the number of Schwann cells in the co-cultures with MSCs while maintaining a sufficient level of growth factors in a 3D matrix for neurite extension.

## 2. Results

### 2.1. Selection and Optimization of Hydrogel Matrices

Prior to bioprinting, co-cultures of MSCs with Schwann cells were first established in simple 3D fibrin hydrogels. The principal requirements of the desired fibrin hydrogel were to support the viability of both cell types, have reasonable matrix disintegration over time, and have the overall rigidity of a gel that is easy to handle. To determine this, fibrin hydrogels of increasing concentrations were formed using fibrinogen (mg/mL) to thrombin ratios (IE/mL) of 1.8/10, 4.5/25, 9/50, and 18/100. MSCs and Schwann cells cultured within these hydrogels (1:1 ratio) were assessed first for viability after 72 h. At a volume ratio of 1.8/10 in 0.5 mL of hydrogel, the resultant fibrin was too thin and scaffold degradation too rapid to be used for viability assays at 72 h. Cells cultured with the fibrin ratios 4.5/25 and 9/50, however, had good viability and better gel stability (Figure 1A), lasting up to 2 weeks. Cells cultured in the 18/100 ratio presented a rounded morphology and had a higher degree of cell death despite the fibrin having relatively good stability. As such, the fibrin hydrogel formed using the 9/50 ratio was chosen for use in all subsequent experiments and this gel composition is referred to hereafter in the text as ‘3D substrate’.

After 72 h in culture, the cell morphology in the hydrogels was ascertained by fluorescently staining the MSCs for surface marker CD90 and the Schwann cells for S100 (Figure 1B). In control experiments, Schwann cells and MSCs were cultured on planar tissue culture polystyrene substrates (2D substrate) and stained with further characteristic markers (Appendix A). MSCs showed the greatest difference in morphology, being generally smaller (less spread) and rounded or spindle-like when cultured in the 3D substrate.

To assess cell proliferation, regression curves using reduction measurements of AlamarBlue™ reagents were determined for MSCs and Schwann cells using both 2D and 3D cell culture systems (Figure 1C). Cells were cultured both individually and together (1:1). Regression was ascertained over a longer time frame (10 days) to discern the total metabolic activity of the cell cultures and show that the cells continued to proliferate within the fibrin hydrogels. Within the fibrin hydrogels, MSC showed extended growth after day seven compared to a slowed growth pattern on the 2D substrate at the same time points.

### 2.2. Expression of Growth Factors by Cultured MSCs and Schwann Cells in 3D Substrates

To assess cell functionality after 1 week of culture in 3D substrates, we analyzed gene transcription levels of various neurotrophic and angiogenic growth factors: the brain-derived neurotrophic factor (BDNF), the nerve growth factor (NGF), neurotrophin 3 (NTF3), angiopoietin-1 (ANGPT1), the glial cell line-derived neurotrophic factor (GDNF), and the vascular endothelial growth factor-A (VEGFA) (Figure 2). Transcription levels of Schwann cell and MSC co-cultures (1:1) were ascertained relative to individual cell cultures. MSCs are used as a baseline (expression level held nominally at 1) against which activity is measured. Total cell numbers were kept the same for all three test groups (1 × 10^5^ cells), and thus SC-50; the expected value for half the SC population in co-culture (1:1) is marked on the graphs as a dashed line. This allows easier visualization of the effect of MSCs in the co-culture set.

Gene transcription levels were higher for all six factors in either co-cultures or Schwann cells alone when compared with levels in MSCs. For co-cultures, however, the expression levels were more varied. Levels of BDNF, NTF3, and GDNF showed increases that were comparable with Schwann cells (*p* > 0.05). Although showing higher levels compared to MSCs, NGF, and VEGFA levels were significantly less than the Schwann cell sample set, as well as the expected values for SC-50, indicating a downregulation of these genes. Lastly, ANGPT1 did not show any effect of co-culture, with the SC-50 experimental value closely matching the expected value.

These initial results suggested that the co-culture of Schwann cells with MSCs produced a selective effect on growth factor expression profiles. To confirm the gene transcript results, we performed enzyme-linked immunosorbent assays on the corresponding culture medium collected from the samples. In addition, we further expanded the co-culture compositions to include lower (1:3) and higher (3:1) ratios of Schwann cells to MSCs to ascertain whether these cell numbers produced a population-dependent effect on growth factor production or if a desirable effect could be achieved at a threshold population. The results from the ELISA assays did not follow those observed at the transcriptional level, suggesting post-translational modification or that these compounds are not completely excreted extracellularly (Figure 3). There were no statistically significant differences between sample sets, suggesting that continued production of growth factors into the culture system by Schwann cells alone could be also maintained at the same level by adding MSC at different concentrations.

### 2.3. Effects of 3D-Printed Cells in Fibrin Hydrogel on Neurite Outgrowth

To evaluate the effects of different ratios of Schwann cells to MSCs on neurite outgrowth from adult DRG neurons, we used a 3D-printed track design to model the 3D-organization of the matrix and cells in a nerve conduit. To achieve high spatial resolution (<300 µm) of our 3D-printed matrix and at the same time keep cell damage low, we used a 260 µm tapered nozzle (see Appendix A). One day after printing, the Schwann cells and MSCs had a rounded morphology but within 96 h they both adopted a more spread morphology (Appendix A). Our model also enabled us to compare the effects of orientated versus randomly placed MSCs and Schwann cells on neurite outgrowth (Figure 4 and Appendix A).

Both the 2D and 3D cell-free substrates showed the shortest mean length of neurite outgrowths (104.5 ± 80 µm and 67 ± 27 µm, respectively). First, we analyzed the overall neurite lengths without segregation into regions (Appendix A) and found very little difference between the substrates, as they equally show a short range and statistically insignificantly increase in lengths when the Schwann cell populations are increased from 25% to 75% (1:3–3:1). Although the pure Schwann cell population had an altogether higher mean value for both the 2D and 3D substrate (668 ± 219 µm and 522.6 ± 230 µm respectively), large variations within the sample set resulted in non-significant differences from the co-culture groups.

Following this, the orientation of the DRG neurites was determined by ascertaining the angle or degree at which the distal end of the neurite shifts relative to the cell soma (Figure 4A). Angles measured within a 45° shift (region A) are noted as remaining within the bioprinted tracks, while those outside this boundary (region B) do not, and likely cross between tracks.

Orientation measurements, however, showed that neurites tended to stay along the printed tracks when the Schwann cell population was less than 100%. The total population that remains within region A was from 60% for 3:1 to 100% when only MSCs (no Schwann cells) were used for bioprinting (Figure 4B). When compared to the non-printed or 2D substrates, the neurite numbers in region A were altogether higher for the 3D-printed groups with the exception of the Schwann cell sample group. Bioprinting with only Schwann cells gave neurite numbers that were comparable to that ascertained for its equivalent 2D substrate where the population for the 2D and 3D substrates were 58.3 and 57.1%, respectively.

Measurements of primary neurite lengths on the 2D substrate showed no significant difference between region A and region B, having on average comparable extension lengths (Figure 4B). In contrast, a difference in behavior is noted for the bioprinted 3D substrates where the longer neurite extensions are restricted to region A and, therefore, occur primarily along the printed tracks. Interestingly, this restriction is only significant when the Schwann cell numbers are reduced showing the biggest difference between the printed MSCs and SC:MSC co-culture ratios 1:3 and 1:1 (Figure 4B).

## 3. Discussion

Three dimensional bioprinting can be used to facilitate complex in vitro assays, as well as the bio-fabrication of nervous tissue-like constructs for in vivo transplantation. Thus, its use and relevance in biomedical research and clinical applications are very promising. In this study, we explored the fluid properties of fibrin hydrogels used as a bioink in extrusion-based 3D bioprinting to optimize the number of viable cultured cells obtained during the printing process and to provide guided neurite outgrowth from adult sensory neurons. Bioinks themselves need to possess several suitably optimized characteristics for the 3D printing process. These include shear stress, viscosity, and gelation. For combination materials, these properties need to be complementary to supporting cell viability attachment and proliferation. Due to the high number of available bioinks (natural and synthetic), as well as the cell types within the body, a one size fits all approach to methodology is not plausible. Previous studies have shown that cell viability and functionality are based on several factors inclusive of the cell type, where a set of parameters using a particular bioink may not be suitable for a different type of cell. This is particularly true of cells with a high sensibility, such as stem cells [22,26,27].

The fibrin hydrogel used in the present study is a well-known matrix to deliver cultured glial and stem cells to the injury sites in both the central and peripheral nervous systems [28,29,30,31,32]. Since pressure and shear stress during printing can negatively affect cell viability [33,34,35], there is a need for the optimization of printing parameters, such as the shape of the syringe and the applied pressure to preserve cell viability during the printing process (see also Appendix A, section “Optimisation of the printing parameters” in Appendix A). Such stresses can kill Schwann cells during the printing process if the hydrostatic pressure is higher than 500 kPa and if cells are exposed to either high shear stress for a short time or moderate shear stress for long times [36,37]. Therefore, to ensure a slow, steady, and controlled flow with minimal cell damage due to the hydrostatic pressure when printing, we set the working pressures to 50 kPa and 10 kPa for the cylindrical and conical nozzles, respectively. These are significantly lower than the cell viability threshold of 500 kPa. To investigate the pressure and shear stress in our experimental setup, we used numerical fluid simulations, which enabled us to choose a syringe that gave us good spatial resolution and a high volume flow rate without compromising cell viability. This allowed us to further explore the combination of printed co-culture systems (MSCs and Schwann cells) and biomaterials to aid research into future nerve conduit design.

During the initial selection and optimization of hydrogel matrices, we found a delayed onset of MSC growth in 3D cultures. This observation is in line with previous studies that have shown that MSCs cultured in fibrin gels have enhanced survival and proliferation properties [38,39,40]. In contrast, Schwann cells had reached confluence by day 5 but exhibited a considerably slowed growth rate in the fibrin gel. This observed difference can be attributed to the fact that fibrin is not specifically treated to promote SC adherence and growth as is the typically Poly-L-lysine-coated tissue culture plastics that are routinely used for two-dimensional culture. The co-cultured set on 2D substrates reflected an intermediate state between the two cell types. In comparison, the fibrin co-culture showed a profile similar to that of the MSCs.

In this study, we assessed the possibility of reducing the number of Schwann cells needed to maintain a sufficient level of growth factors in a 3D matrix. Transcription levels of neurotrophic factors in co-cultures were lower than in the Schwann cells alone. However, analysis of growth factor secretion in the culture medium using ELISA showed that low (1:3) density of Schwann cell populations in MSC co-cultures was observed to be comparable to higher (1:1 and 3:1) densities of Schwann cell populations. Several studies have shown that transcription levels are not necessarily predictive of protein expression levels and this, particularly for neuronal tissue types, is dependent on the class of protein [41]. Other factors inclusive of changes to the cell microenvironment are known to have effects on post-transcriptional activity ranging from translational rate modification to protein half-life and synthesis delay [42]. Although, an intriguing point, the exact mechanisms by which the production of these trophic factors result in an observed equilibrium is beyond the scope of this study and, therefore, a question that will need to be answered within a separate setting. However, the data describes potentially the capability of reducing the number of donor Schwann cells needed for the preparation of nerve conduits.

Another possible explanation could be related to the specific properties of the MSCs. These cells are known to adopt a state of overall quiescence or low activity, which subsequently undergoes evident alteration when presented with chemical, physical, or topographical cues [43]. In this case, being cultured in the presence of Schwann cells inevitably causes a mirroring effect in MSCs, as they undergo phenotypical changes increasing the production of growth-promoting factors [14,44,45]. It is noted that in the 3D substrate, the average measured neurite length at a high Schwann cell (3:1) concentration is slightly lower than the other concentrations. This, however, is not a significant change and is primarily brought on through variation in the sample set. Although MSCs have not been conclusively shown to undergo terminal differentiation to neurons or glial cells, they instead develop into cells mimicking a number of neuronal and glial cell characteristics, including the expression of various growth factors [46,47]. These cells are able to promote neurite outgrowth [48] and, when used in co-culture with Schwann cells, improved therapeutic effects in injury regeneration, an effect that is not observed when MSCs alone are used [49]. The use of undifferentiated MSCs in this experimental setup is also advantageous as it removes the need for the added step of differentiation ex vivo prior to use. The use of adult MSCs in this co-culture system is of particular benefit as they possess less plasticity compared to other stem cell types. Embryonic stem cells, for example, run the risk of spontaneous tumor formation without first undergoing directed differentiation prior to use in vivo [50].

Similar studies by Ning et al. [51] have shown that dispensing properties during the printing process allows the alignment of fibrin fibers in the direction of printing. While this does not necessarily orientate the printed Schwann cells, a finding also confirmed in this study and in addition is also the case for MSCs, it does facilitate the alignment of DRG neurites. No significant effects were noted regarding the overall lengths of the primary neurites between both the co-culture test groups and the 2D and 3D substrates (Appendix A).

While the comparison between the 2D and 3D substrates showed no difference in primary neurite lengths, the presence of Schwann cells invariably affected neurite lengths, growing from an average of 289.5 µm when they made up 25% of the population, to 600 µm when Schwann cells were 100%. Therefore, the total Schwann cell population, in conjunction with spatial restriction or printing parameters, needs to be carefully balanced to achieve directed outgrowth of neurites. The use of over 75% Schwann cells resulted in lessened directional outgrowth (Figure 4C). A reason is that at these numbers, cell–cell interactions are not sufficiently dampened across the printing distances to prevent adjacent neurite crossovers from occurring. At lower numbers, this is not the case and a significant restriction to the printed tracks is retained at 50% or fewer Schwann cells (Figure 4B). Given that the average neurite lengths are not significantly changed between 25 and 75% Schwann cells and spatial restriction is maintained, it follows that by combining spatial restriction with Schwann cell co-cultures, the need for high Schwann cell numbers could be circumvented.

Continued research of complementary cell types for use with (or without) Schwann cells in peripheral nerve regeneration is primarily driven by its short supply and relatively long-term expansion time in culture compared to its demand for use in the clinic. As such, attaining the capability to significantly reduce the amount of required Schwann cells is a great advantage. This study can be a premise for ultimately constructing a biosynthetic neural conduit for use in vivo. As there is no significant difference in neurite length between 2D and 3D substrates, the printed fibrin tracks serve primarily to restrict neurite outgrowth and thus promotes directionality. Furthermore, creating a focus on increased directionality, in theory, should also allow axons to traverse the length of the conduit in less time to reach the host Schwann cells in the distal nerve stump and help to maintain their functionality. Interestingly, in a recent clinical trial on spinal cord injury (SCI) patients, the combination of autologous SC and bone marrow MSC transplantation at the subacute stage of SCI revealed that this combination of cells is safe and reduces the need for SC for clinical application [52].

The sensitivity of neurons to their microenvironment requires that biomaterials intended for nervous tissue repair can mimic the natural matrix as closely as possible. In this sense, the biomaterial and the methods used in its creation are undoubtedly of fundamental importance. Several studies have reported on the use of topographic and geometric cues in substrate design for neuronal regeneration. Among this is the use of fibrillar matrices, such as fibrin, which, in addition to having a tunable crosslinking property to populate growing cells, can provide spatial guidance for neuronal growth and extension [53,54].

## 4. Conclusions

In summary, the findings from this study demonstrate the correlation of cell behavior with their microenvironment and a potential means of optimizing axonal guidance across a conduit with a lessened demand for Schwann cells. While still in its very early stages, the study intends to highlight the feasibility of using an instructive 3D cell printed construct onto a polymer base, which can subsequently be converted into a hollow conduit to bridge an injury gap following reconstructive surgery after peripheral nerve injury.

## 5. Materials and Methods

### 5.1. Preparation of Hydrogels

Fibrin hydrogels were made using two-compound fibrin glue (Tisseel^®^ Duo Quick, Baxter, Lund, Sweden) containing 90 mg/mL of fibrinogen, 10 IU of factor XIII, 5.5 mg of fibronectin, 0.08 mg of human plasminogen, and 3.4 UPE of bovine aprotinin to be mixed with 500 IU of human thrombin in 40 mM calcium chloride, 149 mm sodium chloride, 40 mm glycine, and 50 mg/mL human serum albumin. Fibrinogen for all experiments was diluted using basal Dulbecco’s Modified Eagle Medium (DMEM; Invitrogen Life Technologies, Carlsbad, CA, USA), while thrombin was diluted using phosphate-buffered saline (PBS). A total of four fibrin hydrogel compositions were made by crosslinking the fibrinogen (mg/mL) with thrombin (IE/mL) to obtain 0.5 mL of final hydrogel: (i) 1.8/10, (ii) 4.5/25, (iii) 9.0/50, and (iv) 18.0/100.

### 5.2. Cell Cultures

Primary cultures of the rat Schwann cells, rat bone marrow-derived mesenchymal stem cells (MSCs), and adult rat dorsal root ganglion (DRG) sensory neurons have been prepared as described in our previous studies [53,55,56,57]. This study was approved by the Animal Review Board at the Court of Appeal of Northern Norrland in Umeå, Sweden (DNR #A22-15 and #A46-15).

Schwann cells were obtained from the sciatic nerves of adult female Sprague–Dawley rats (n = 5). Nerves were cut into 1–2 mm long pieces and transferred in culture plates in DMEM/10% heat-inactivated fetal calf serum, FCS, and kept at 37 °C, 95% humidity, and 5% CO_2_. After 2 weeks, the nerve pieces were enzymatically dissociated and replated on 25 cm² poly-D-lysine-coated (PDL) tissue culture flasks in DMEM/10% FCS, supplemented with neuregulin NRG1 (R&D Systems) and 10 µM forskolin (Invitrogen). Once the Schwann cells became confluent, the purification procedure was carried out using a medium containing 10 µM cytosine-b-D-arabinofuranoside (Sigma-Aldrich, St. Louis, MO, USA). The purity of cells was assessed by using immunostaining for S100 protein and was approximately 95%.

To isolate and prepare primary cultures of MSCs, the tibia and femur of adult female rats (n = 5) were isolated, washed with normal saline, and the epiphyses were removed. The bone marrow was flushed with the medium using a 21 G needle attached to a 5 mL syringe. After centrifuging the suspension at 1500 rpm for 5 min, the supernatant was removed and the cells were re-suspended in the same medium. The resulting suspension was filtered through a 70 µm nylon mesh and plated on 75 cm^2^ culture flasks. After 24 h, the supernatant containing non-adherent cells was discarded and a fresh medium was added. The medium was changed every 48 h. After 7 days when the cells reached approximately 50% confluence, they were detached with 1.25% trypsin/EDTA and re-plated in culture flasks at a density of 5 × 10^3^ cells per cm^2^. The cells were expanded for additional 2 weeks and then were assessed for expression of characteristic MSC cell surface markers and differentiation capability in vitro (Appendix A). Co-cultures of Schwann cells and MSCs were combined from counted re-suspended cell pellets to give final ratios of 1:3, 1:1, and 3:1. Cells were then cultured using conditions described for Schwann cells alone.

Cultures of DRG neurons were prepared from adult female rats (n = 3). The C2-L6 DRGs were removed bilaterally and transferred into ice-cold Neurobasal™-A medium (Invitrogen Life technologies). DRGs were then enzymatically dissociated using 0.125% collagenase Type IV (Worthington Biochemicals Corp., Lakewood, NJ, USA) and 0.25% trypsin (Worthington), washed, and further dissociated by gentle trituration. Cells were centrifuged with 15% bovine serum albumin at 100g for 10 min to remove most non-neuronal cells. To achieve approximately 95% purity, this process was repeated, and neurons were then resuspended in Neurobasal™-A medium with B-27 supplement and 0.5 mM l-glutamine (Sigma Aldrich) and seeded onto coverslips pre-coated with 2 µg/mL laminin-1 (Sigma-Aldrich) as controls or onto 3D-printed hydrogels.

For the initial testing of bioprinting parameters, we used human MSCs isolated and characterized as described in our previous studies [55,58,59]. Written informed consent was obtained from donors. Collection, culture, storage, and usage of all clinical isolates were approved by the local ethics committee for research at Umeå University (Dnr 2013-276-31M).

### 5.3. Immunocytochemistry

Cultured cells were fixed in 4% (*w*/*v*) paraformaldehyde for 20 min at room temperature. After blocking with 5% normal horse and goat serum in 0.1% BSA, 0.1% sodium azide, and 0.1% Triton for 15 min, the primary antibody was applied for 2 h at room temperature. The following primary antibodies were used: mouse anti-CD34 (1:200; Chemicon), mouse anti-CD45 (1:200; Chemicon), mouse anti-CD63 (1:200; Chemicon), mouse anti-CD73 (1:200; Chemicon), mouse anti-CD105 (1:200; Chemicon), mouse anti-CD146 (1:200; Chemicon), mouse anti-Thy1.1 (CD90; (1:1000; Chemicon), and rabbit anti-S-100 protein (1:1000; Dako). Cells were then washed with PBS and incubated with normal serum followed by fluorescently conjugated secondary antibodies (Alexa Fluor 488 or 568, 1:300 dilution; Molecular Probes, Invitrogen) for 1 h in the dark. Cells were mounted with ProLong mounting media containing 4′-6-diamido-2-phenylindole (DAPI; Invitrogen Life Technologies).

### 5.4. Live/Dead Cell Assay

Cell viability was determined using 2 μM calcein AM and 4 μM ethidium homodimer-1 (Invitrogen Life Technologies) in PBS. The solution was added to the culture well at 10% of the final volume and incubated at 37 °C for 30 min. Samples were subsequently fixed in 4% paraformaldehyde in phosphate buffer (pH 7.4) for 20 min at room temperature, washed, and placed in PBS prior to fluorescence microscopy. Assays were performed at 72 h for free-form fibrin hydrogels and at 24 h for 3D-printed samples (see Section 5.8).

### 5.5. Cell Proliferation Assay

AlamarBlue^TM^ (Thermo Fisher Scientific, Waltham, MA, USA) was used to assess the effects of cell co-culture and the fibrin hydrogel on cell proliferation. At each time point (1, 3, 5, 7, and 10 days in culture) AlamarBlue™ (100 µL per 1 mL media) was added for 3 h. Following this, a 200 µL solution was transferred into a 96-well plate, and the optical density was determined by measuring the absorbance at 570 nm and with correction at 600 nm using a Synergy HT microplate reader with Gen5 software v2.04 (Biotek Instruments, Santa Clara, CA, USA). The media in each well was changed to fresh medium and the cells were measured in the same manner at the next determined time point. Each measurement was performed using four replicate samples.

### 5.6. RT-PCR

Total RNA was isolated from the cells cultured within fibrin after 1 week using an RNeasyTM kit (Qiagen, Hilden, Germany). Culture media supernatant was removed from each well plate containing the fibrin hydrogels and this was saved for subsequent ELISA assay (Section 5.7). A small amount of media was left in the culture well plate. The hydrogel was disintegrated into the media by gentle mechanical dissociation and a lysis buffer was then added to the mixture, and dissociation continued. Subsequent RNA isolation was carried out as per the manufacturers’ protocol. The One-Step RT-PCR kit (Qiagen, Hilden, Germany) was used for all RT-PCR, as per the manufacturer’s instructions, with the addition of 1 ng total cellular RNA. RT-qPCR was performed using SsoFastEvaGreen supermix (Bio-Rad, Hercules, CA, USA) in a CFX96 Optical Cycler and analyzed using the CFX96 manager software (Bio-Rad). Reactions were optimized and processed according to the manufacturer: For qRT-PCR, 10 ng/μl reaction mix was converted into cDNA using an iScript™ cDNA synthesis kit (Bio-Rad). qRT-PCR was performed using SsoFast™EvaGreen supermix (Bio-Rad) in a CFX96 Optical Cycler and analyzed using the CFX96 manager software (Bio-Rad). Reactions were optimized and processed according to the manufacturer with initial denaturation/DNA polymerase activation at 95 °C for 30 s followed by PCR: 95 °C for 5 s, variable annealing temperature (see Table 1) for 5 s, and 65 °C for 5 s repeated for 40 cycles. Expression levels were normalized to the reference genes GAPDH. Quantification analysis was performed using the comparative ΔΔCt method [60] and gene expression was expressed as fold change relative to the control sample. Samples were assayed using six replicate samples and gene expression was expressed as mean ± SEM.

### 5.7. Enzyme-Linked Immunosorbent Assays

Enzyme-linked immunosorbent assays were performed on cell medium using RayBio^®^ Sandwich enzyme-linked immunosorbent assay kits (RayBiotech, Inc.， Peachtree Corners, GA, USA) as per the manufacturer’s instructions.

### 5.8. Preparation of Hydrogels and Cells for 3D Bioprinting

#### 5.8.1. Numerical Simulations

The extrusion printing technique applies pressure using a piston (Appendix A); therefore, the printed hydrogel is affected by hydrostatic pressure and shear stress from the friction inside the material, as well as between the material and surfaces inside the syringe. To estimate the shear stress distribution inside the printing syringe and nozzle we carried out numerical simulations of the biomaterial flow. Since the hydrogel exhibits non-Newtonian properties, implying that the viscosity changes with its fluid velocity, we included this in the physical model, which we solved using COMSOL Multiphysics, version 5.5.

To find the velocities of the hydrogel in the syringe, we numerically solved the full stationary Navier–Stokes equations, described by:(1)$$\rho u\cdot \nabla u=-\nabla p+\eta \nabla^{2}u,$$
where ρ is the fluid density, **u** is the velocity vector, and p is the pressure. At the syringe inlet, we used a parabolic velocity profile with a center velocity that gives the flow rate specified by Appendix A. At the outlet, the pressure was set to zero and on all surfaces we applied the no-slip condition, **u** = 0. Finally, we set the initial fluid velocity in the syringe to zero.

The software estimates the fluid forces by numerical integration of the fluid shear stress and pressure, using the solution given by Equation (1) (Appendix A). To validate that our simulations produced reliable data, we compared our results with those in [27]. In that work, they compared analytical solutions for cylindrical and conical nozzles with experimental data, which showed good agreement. Thus, we could, in turn, validate our pressure and shear stress simulations directly with their results.

#### 5.8.2. Experimental—Initial Testing Parameters

To evaluate possible cell damage during 3D printing, we first printed cells in GelMA (Gelatin Methacrylated) hydrogel (CELLINK AB; Gothenburg, Sweden) using either a 200 µm diameter cylindrical nozzle with a length of 12.7 mm or a 410 µm tapered (conical) nozzle with a length of 20.0 mm. Printing was performed using a pneumatic extrusion Inkredible printer and accompanying software (CELLINK AB). Instead of printing shapes, the 3D constructs were printed directly into 1.5 mL Eppendorf tubes. For each sample, a defined volume (25 µL) of the 3D-printed output (35,000 cells) was used. The 3D constructs were cross-linked using UV light (405 nm LED) for 5–10 min. As controls, cells were also cultured on planar surfaces (2D control) and within GelMA gels (3D control, non-printed). Polymerized 3D constructs were subsequently digested in a solution of 3 mg/mL Collagenase type I (Worthington Biochemicals Corp., Freehold, NJ, USA) and 4 mg/mL Dispase II (Roche Diagnostic/Boehringer Mannheim Corp., Indianapolis, IN, USA) in a 37 °C water bath for 60 min. Every 10 min, the tubes were vortexed to help dissolve the hydrogels. The cells were pelleted and washed twice by centrifugation at 400 g for 5 min and then re-suspended in PBS. The cells were then subsequently stained for viability using a live/dead assay kit. After 20 min, the cells were pelleted by centrifugation at 400 g for 5 min and washed with PBS. The cells were then resuspended in 500 µL PBS. Cell population data were acquired by flow cytometry, BD Accuri™ C6 Plus (BD Bioscience). The viability of 5000 cells (events) was analyzed with flow cytometry (Accuri™ C6 Plus, BD Bioscience). Results showed no statistically significant differences in the total number of cells recovered or viability across the conditions tested (Appendix A).

#### 5.8.3. Experimental—Final Parameters for Fibrin Hydrogel

The ‘bioink’ for printing was based on fibrin hydrogel consisting of a 9 mg/mL fibrinogen solution, as described in Section 2.1, thus using the same 9/50 ratio as per the selection determined by the results of experiments shown in Figure 1A,B. Prior to printing, fibrinogen was allowed to flow through the nozzles to prime them. The fibrinogen solution was printed onto the substrate (see Section 5.9) using a 20 mm 260 µm bore tapered nozzle and a working pressure range of 6–10 kPa to optimize printing resolution and minimize cell damage. A total of 1.0 × 10^6^ cells suspended in 1 mL of fibrinogen were used. A singular printed track consisted of approximately 10–15 µL of fibrinogen. For subsequent experiments, fibrin tracks were fabricated as follows: (i) cell-free fibrin, (ii) containing MSCs only, (iii) a 3:1 MSC to Schwann cell ratio, (iv) a 1:1 MSC to Schwann cell ratio, (v) a 1:3 MSC to Schwann cell ratio, and (vi) Schwann cells only. Cells in “free form” fibrin gel served as baseline controls.

### 5.9. PCL Substrates for 3D Printing of Hydrogels

Weighed compression was used to create flat polycaprolactone (ElogioAM Haarlem, Netherlands) substrates approximately 5 cm in length and 2 cm wide. These were washed in 70% ethanol, allowed to air dry for 20 min, placed into thrombin solution (50 IE/mL), and adsorption was allowed to take place overnight at 37 °C. Following this, substrates were removed from the thrombin solution and allowed to air dry completely (48 hrs), after which they were then used as the bioprinting substrates. The fibrinogen solutions were printed onto thrombin-coated PCL substrates in a series of lines (tracks) measuring 50.0 mm in length, 5.0 mm wide, and 0.1 mm high. Crosslinking was initiated on contact with the thrombin surface and allowed to complete for 2 min at room temperature before samples were placed into culture media and maintained at 37 °C and 5% CO_2_ for the duration of the experiment. After printing, substrates were used only up to 48 h for DRG to avoid any artifacts that may arise from fibrinogen degradation.

### 5.10. Neurite Outgrowth Assays

DRG neurons isolated from adult rats were plated (3000 per cm^2^) onto 3D-printed fibrin tracks containing different Schwann cell and MSC populations as described above. Cells were allowed to settle and attach for 1 h and culture media was then added. The cultures were incubated for 48 h at 37 °C and 5% CO_2_. DRG neurons seeded on poly-D-lysine coated coverslips were also incubated in conditioned media for 48 h (2D or non-printed substrate). Cells were then processed for immunocytochemistry using mouse anti- βIII tubulin antibody (1:200; Sigma Aldrich) and the length of neurites was measured as previously described [11]. Orientation and Sholl analysis were also performed. Immunofluorescent microscopy images of DRG neurons stained for βIII-tubulin were converted into 8-bit tracing images (thresholded 8-bit image) in ImageJ. Neurite orientations were determined by measuring the angle between the distal end of the primary neurite and the center of the cell soma aligned along the length of the printed track. Angles (δ) measured within 45° were grouped into region A and those measuring outwith (greater than 45° and up to 90°) were grouped into region B (Figure 4A). Sholl analysis was then carried out using the Fiji plugin in ImageJ [61]. Analyses were carried out using a 150 µm maximum radius from the cell soma and 10 µm intervals (Appendix A).

### 5.11. Statistical Analysis

Analysis of variance (ANOVA) and Tukey’s post hoc tests were performed using GraphPad Prism software to determine statistical differences between the experimental groups (Prism^®®^, GraphPad Software, Inc; San Diego, CA, USA). Statistical significance is noted where the calculated *p*-value is less than 0.05 using a minimum of four biological replicates, unless otherwise stated.

## Figures and Tables

**Figure 1 gels-09-00172-f001:**
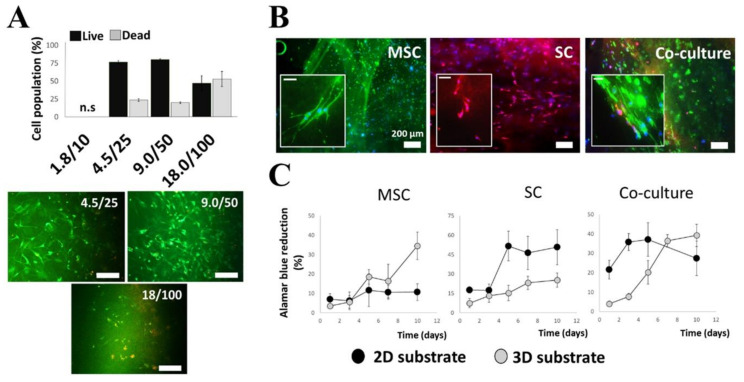
(**A**) Cell viability: Schwann cells (SCs) and MSCs were co-cultured (1:1) in fibrin hydrogels using various fibrinogen/thrombin ratios to alter resultant gel stiffness. The softest hydrogel is 1.8/10 and the stiffest is 18/100. Viable populations were best for cells within 4.5/25 and 9.0/50. (**B**) Immunofluorescence staining of SCs, MSCs, and co-cultured cells within 9.0/50 fibrin (3D) substrates to ascertain cell morphology. SCs were stained for S100 (red), MSCs for CD90 (green), and cell nuclei with DAPI (blue). (**C**) AlamarBlue reduction profiles to ascertain comparative cell survival and proliferation using 2D and 3D substrates. Error bars are standard deviations from the mean; n = 6; scale bars are 100 µm except where stated otherwise and are 50 µm for image inserts in (**B**).

**Figure 2 gels-09-00172-f002:**
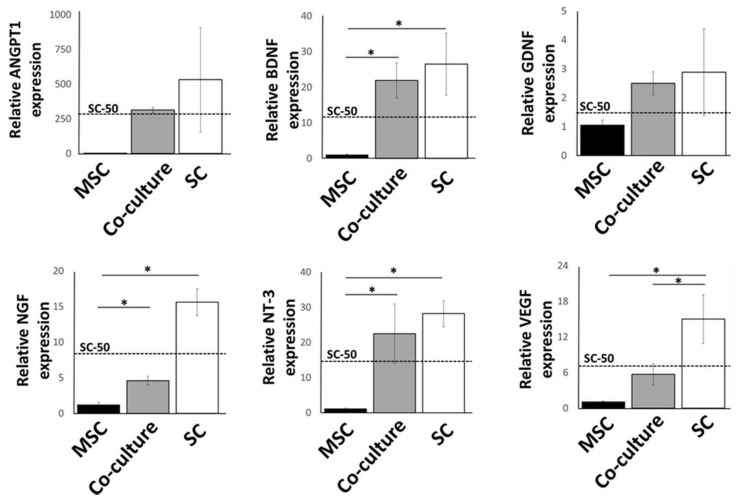
qRT-PCR analysis of neurotrophic and growth factor expression in MSCs, Schwann cells (SC), and co-cultured cells (1:1). Relative expression levels are given with respect to the MSCs samples with expression value set nominally at 1. Dashed line shows SC-50; the expected value at 50% Schwann cell population. Error bars are standard errors from the mean; * indicates a significant difference between groups as determined by one-way ANOVA followed by Tukey’s post hoc test where *p* < 0.05; n = 4.

**Figure 3 gels-09-00172-f003:**
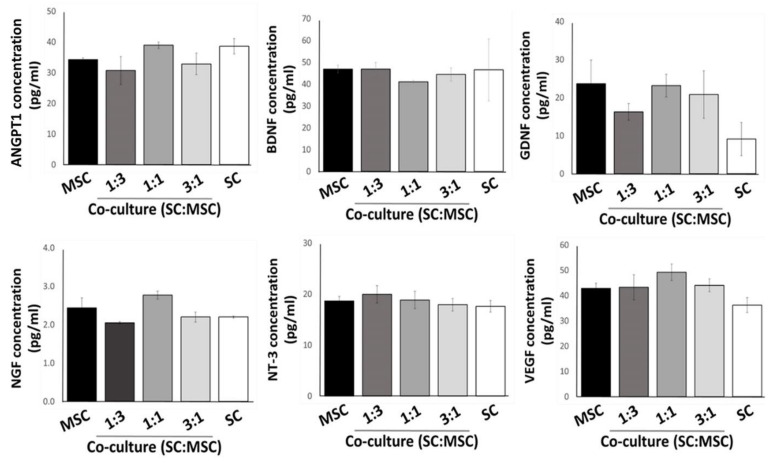
ELISA analysis of culture media samples taken from MSCs, Schwann cells (SC), and varied co-culture compositions of both cell types. Error bars are standard errors from the mean, n = 4.

**Figure 4 gels-09-00172-f004:**
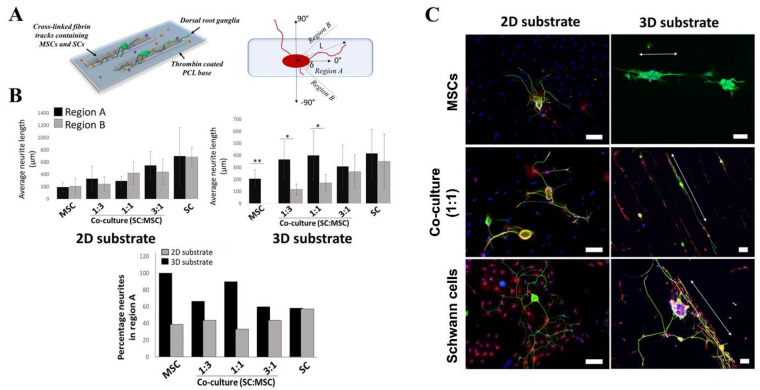
(**A**) Schematic illustrating printing of the 3D fibrin substrate onto PCL and orientation measurements: the angle (δ) was measured between the cell soma center and the distal end of the neurite. (**B**) Average lengths of primary DRG neurites on flat (2D) and printed (3D) substrates. Lengths are shown for neurites that fall within region A (δ up to 45°) and neurites that fall within region B (δ between 45 and 90°). (**C**) Fluorescence staining of DRG neurons and Schwann cells (SCs). SCs were stained for S100 (red) and used to highlight spatial organization and the population of printed cells. DRG neurons were stained for βIII-tubulin (green) and cell nuclei with DAPI (blue). The white arrows in the 3D substrate column indicate the direction of the printed fibrin track. Error bars are standard deviations from the mean; * indicates a significant difference between groups where *p* < 0.05, ** *p* < 0.01; n ≥ 10. All scale bars are 100 µm.

**Table 1 gels-09-00172-t001:** PCR primers used to quantify mRNA expression.

Gene Name	Sequence:	5′ → 3′	Annealing Temperatures (°C)
ANGPT1	Forward Reverse	GAA AAT TAT ACT CAG TGG CTG GAA AA TTC TAG GAT TTT ATG CTC TAA TAA ACT	58.4
BDNF	Forward Reverse	ATG GGA CTC TGG AGA GCG TGA A CGC CAG CCA ATT CTC TTT TTG C	66.9
GDNF	Forward Reverse	GGG TTT AGC TTT CAA GGG CTG A CGGAACAGACAGACAAATGG	61.8
NGF	Forward Reverse	AAG GAT CCT GGA CCC AAG CTC ACC TCA GAG TGA CGT GGA TGA GCG CTT GCT CCT	74.1
NTF3	Forward Reverse	CTT ATC TCC GTG GCA TCC AAG G TCT GAA GTC AGT GCT CGG ACG T	65.6
VEGFA	Forward Reverse	TGC ACC CAC GAC AGA AGG GGA TCA CCG CCT TGG CTT GTC ACA	70.9
GAPDH	Forward Reverse	CCC CCA ATG TAT CCG TTG TG TAG CCC AGG ATG CCC TTT A	63.3

## Data Availability

All datasets generated and/or analyzed during the current study are available from the corresponding author upon reasonable request.

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
