# Peer review of "Bioprinted Schwann and Mesenchymal Stem Cell Co-Cultures for Enhanced Spatial Control of Neurite Outgrowth"

_gels, 2023, doi:10.3390/gels9030172_

Round 1
Reviewer 1 Report
The presented study reports the fibrin hydrogels fabricated by 3D bioprinter supplemented with cultured mesenchymal stem cells (MSCs) and Schwann cells to assess the cell viability, production of neurotrophic factors and the ability of the printed cell matrix to spatially control neurite outgrowth from the adult sensory dorsal root ganglion neurons. Though interesting results are obtained, the novelty of this study is poor since no related published work in recent five years is cited and discussed, thus needing to be well enhanced. Besides, some other issues should be also further addressed:
In Figure S1, the scale of Figure B is not illustrated; in Figure S1, Figure C is not illustrated; in Figure S1, the change in the expression of CD90 and CD105 with increasing time in Figure D is not illustrated.
Line116-120, Line 161-170 should be moved to the section of Discussion.
Statistical analysis should be performed in Fig.3 and Fig.S2D,E.
The present work should be discussed with the previous published work in Acta Biomater. 2017 Jun;55:296-309. doi: 10.1016/j.actbio.2017.04.010. Epub 2017 Apr 12. The advantages and disadvantages of this work compared with other works should be well discussed.
Author Response
Reviewer #1
The presented study reports the fibrin hydrogels fabricated by 3D bioprinter supplemented with cultured mesenchymal stem cells (MSCs) and Schwann cells to assess the cell viability, production of neurotrophic factors and the ability of the printed cell matrix to spatially control neurite outgrowth from the adult sensory dorsal root ganglion neurons. Though interesting results are obtained, the novelty of this study is poor since no related published work in recent five years is cited and discussed, thus needing to be well enhanced. Besides, some other issues should be also further addressed:
- In Figure S1, the scale of Figure B is not illustrated; in Figure S1, Figure C is not illustrated; in Figure S1, the change in the expression of CD90 and CD105 with increasing time in Figure D is not illustrated.
Response: In the Figure Legend for Figure S1 we wrote: “Scale bars are 100 μm except where stated otherwise.” We also indicated in the Figure Legend that “the expression of CD90 and CD105 was unchanged over time”.
- Line116-120, Line 161-170 should be moved to the section of Discussion.
Response: The text on lines 116-120 and 161-170 (original submission) has been moved to the Discussion.
- Statistical analysis should be performed in Fig.3 and Fig.S2D,E.
Response: Statistical analysis has been performed using ANOVA and Tukey post hoc test as described in section “4.11. Statistical analysis”. For the results shown in Figure 3 we wrote “There were no statistically significant differences between sample sets suggesting that continued production of growth factors into the culture system by Schwann cells alone could be also maintained at the same level by adding MSC at different concentrations.” (please, see lines 158-161 in the original submission). Regarding Fig. S2D and S2E, we wrote that “Results showed that there were no statistically significant differences in total numbers of cells recovered or viability across the conditions tested (Supplementary Figure S2D and E). Please, see lines 479-481 in the original text before revision.
- The present work should be discussed with the previous published work in Acta Biomater. 2017 Jun;55:296-309. doi: 10.1016/j.actbio.2017.04.010. Epub 2017 Apr 12. The advantages and disadvantages of this work compared with other works should be well discussed.
Response: The suggested paper has been added to the Discussion.
Reviewer 2 Report
In this manuscript, the authors have explored bioprinting of MSC and Schwann cell together to facilitate guided axonal regeneration. The manuscript is well-written and the results are well-explained. The reviewer believes this is a paper of high quality and it is suggested that the article be accepted after the following minor comments are taken care of.
1. The abstract needs to be rewritten by adding some numeric data.
2. In the introduction section please elaborate on the need for a neural implant.
3. Please include the image of the bioprinted hydrogel and the distribution of the cells within the hydrogel.
4. Mechanical properties of the hydrogel play a critical role in MSC functionality, please indicate the mechanical property of the scaffold.
5. The author mentioned the scaffold lasted for about 2 weeks. What is the optimal time duration of hydrogel availability for neural regeneration and if this particular hydrogel sustains that long?
Author Response
Reviewer #2
In this manuscript, the authors have explored bioprinting of MSC and Schwann cell together to facilitate guided axonal regeneration. The manuscript is well-written and the results are well-explained. The reviewer believes this is a paper of high quality and it is suggested that the article be accepted after the following minor comments are taken care of.
- The abstract needs to be rewritten by adding some numeric data.
Response: The Abstract has been revised.
- In the introduction section please elaborate on the need for a neural implant.
Response: The Introduction has been revised.
- Please include the image of the bioprinted hydrogel and the distribution of the cells within the hydrogel.
Response: The Figure S4 demonstrates distribution of the printed MSC and Schwann cells in the gels at 24 and 96 hours post printing. We also have corrected our mistake on page 5, line 182 (original submission) where we change “Supplementary Figure S3” to ““Supplementary Figure S4”.
- Mechanical properties of the hydrogel play a critical role in MSC functionality, please indicate the mechanical property of the scaffold.
Response: Unfortunately, we had no technical possibility to measure mechanical properties of the gels in this study. However, recent paper in the Acta Biomaterialia suggested by the Reviewer “1 provides some data on fibrin gel elasticity. The paper has been added to the Discussion as suggested by Reviewer #1.
- The author mentioned the scaffold lasted for about 2 weeks. What is the optimal time duration of hydrogel availability for neural regeneration and if this particular hydrogel sustains that long?
Response: Several research groups used fibrin gels in both peripheral nerve and spinal cord injury. It has been found that hydrogel could remain in the injury site for 1-2 weeks after implantation. This is usually sufficient for delivery of cultured glial and stem cells into spinal cord cavity since the majority of cell (almost 90%) die withing the first 2 weeks after transplantation. Regarding peripheral nerve, the results including our own data , demonstrate that the gel’s stability is sufficient to promote regeneration across a short 10-20 mm nerve defects. However, it is most probably not applicable for the long nerve gaps (40-80 mm) found after injury to the large peripheral nerves and brachial plexus in human patients.
Reviewer 3 Report
The manuscript entitled "BIOPRINTED SCHWANN AND MESENCHYMAL STEM CELL CO-CULTURES FOR ENHANCED SPATIAL CONTROL OF NEURITE OUTGROWTH" reported by Alakpa et al. is interesting, since it brings valuable data to bioprinting techniques that can be used for the development of biosynthetic conduits and characterize Schwann and mesenchymal stem cell co-cultures for enhanced spatial control of neurite outgrowth.
Author Response
- The manuscript entitled "BIOPRINTED SCHWANN AND MESENCHYMAL STEM CELL CO-CULTURES FOR ENHANCED SPATIAL CONTROL OF NEURITE OUTGROWTH" reported by Alakpa et al. is interesting, since it brings valuable data to bioprinting techniques that can be used for the development of biosynthetic conduits and characterize Schwann and mesenchymal stem cell co-cultures for enhanced spatial control of neurite outgrowth.
Response: We thank the Reviewer for these comments.
Reviewer 4 Report
The amount of Schwann cells required to maintain an adequate level of growth factors in a 3D matrix has been successfully determined by the authors. Schwann cell populations in MSC co-cultures were shown to have low (1:3) densities that were equivalent to greater (1:1 and 3:1) densities based on an analysis of growth factor secretion. The goal of the work, as the author stated, is currently in its very early phases, is to demonstrate the viability of employing an instructional 3D cell-printed construct onto a polymer basis that can then be transformed into a hollow conduit. This study may pave ways for other biomaterials to be employed and in future, we may have live nerve conduit by using bioprinting. However, some minor points need to be addressed.
There are references needed for lines 50–62, 228–241, and 288–317.
Authors may please rewrite the introduction's last paragraph (lines 70–72) by using the discussion part (lines 257–258).
In the caption of supplemental Figure S1 A and B, please clearly mention the cells used, i.e., BMSCs or SCs.
Please point out Figure 1C for the discussion of the regression curves using reduction measurements of Ala- marBlue™ (lines 106–120). Also, why does the coculture present a similar trend to MSC in 3D fibrin hydrogel? It would be interesting to know more about the literature supporting this conclusion for the co-culture of MSCs and SCs in hydrogels or printed constructs.
Line 243–256: While discussing numerical fluid simulations, please clearly point out the supporting information for the reader to have a look if needed.
The authors have discussed the significance of research from lines 257–280; they mentioned the first goal, but there are no other goals or aims, so it would be easier to read if this discussion were not quantified, as there is only one goal discussed.
Is it possible to use GelMA or any other fiber materials for the same purpose? Why do the authors specifically choose fibrin glue for this purpose?
Figures S1-3 are mentioned in the paper as Supplementary Figure Sx, whereas Figure S4 cannot be found, while S5 is written as Supplementary Figure 5B, and S6 is not mentioned in the manuscript. To make the setting more comprehensible and easy to read, the authors must homogenize it.
Minor issues;
Figure, figure
Extra spaces between words
Avoid using sentences like "This observation may be in some part due to the presence of MSCs."
Do the authors refer to Figure 4A in line 522?
The authors have ended the discussion with the statement, "This biomaterial can then be used as a biosynthetic conduit to promote axonal regenration after peripheral nerve injury." While this statement is not entirely appropriate and relevant to the findings, it is advised to rewrite it as "exclude" to avoid misunderstandings.
There is no conclusion provided.
Author Response
Reviewer #4
The amount of Schwann cells required to maintain an adequate level of growth factors in a 3D matrix has been successfully determined by the authors. Schwann cell populations in MSC co-cultures were shown to have low (1:3) densities that were equivalent to greater (1:1 and 3:1) densities based on an analysis of growth factor secretion. The goal of the work, as the author stated, is currently in its very early phases, is to demonstrate the viability of employing an instructional 3D cell-printed construct onto a polymer basis that can then be transformed into a hollow conduit. This study may pave ways for other biomaterials to be employed and in future, we may have live nerve conduit by using bioprinting. However, some minor points need to be addressed.
- There are references needed for lines 50–62, 228–241, and 288–317.
Response: We added new references on bioprinting in Introduction and Discussion.
- Authors may please rewrite the introduction's last paragraph (lines 70–72) by using the discussion part (lines 257–258).
Response: The Introduction’s last paragraph has been revised.
- In the caption of supplemental Figure S1 A and B, please clearly mention the cells used, i.e., BMSCs or SCs.
Response: The caption to the Figure S1 A and B has been revised.
- Please point out Figure 1C for the discussion of the regression curves using reduction measurements of AlamarBlue™ (lines 106–120). Also, why does the coculture present a similar trend to MSC in 3D fibrin hydrogel? It would be interesting to know more about the literature supporting this conclusion for the co-culture of MSCs and SCs in hydrogels or printed constructs.
Response: The reference to Figure 1C has been added. We also wrote that “This observation is in line with previous studies that have shown that MSCs cultured in fibrin gels have enhanced survival and proliferation properties [29, 30, 31]” (please, see lines 112-113 in the original submission).
- Line 243–256: While discussing numerical fluid simulations, please clearly point out the supporting information for the reader to have a look if needed.
Response: The reference to the Supplementary information, section “Optimisation of the printing parameters” with Figures S2 and S3 has been introduced in the text.
- The authors have discussed the significance of research from lines 257–280; they mentioned the first goal, but there are no other goals or aims, so it would be easier to read if this discussion were not quantified, as there is only one goal discussed.
Response: The text has been adjusted according to reviewer’s comment.
- Is it possible to use GelMA or any other fiber materials for the same purpose? Why do the authors specifically choose fibrin glue for this purpose?
Response: It has been recently demonstrated that GelMA hydrogel in combination with poly(2- hydroxyethylmethacrylate) can be used as an “outer part of the nerve guide” and it was also reported that Schwann cells attached and proliferated on GelMA (T. Dursun Usal, D. Yucel, V. Hasirci. A novel GelMA-pHEMA hydrogel nerve guide for the treatment of peripheral nerve damages. Int. J. Biol. Macromol., 121 (2019), pp. 699-706).
- Figures S1-3 are mentioned in the paper as Supplementary Figure Sx, whereas Figure S4 cannot be found, while S5 is written as Supplementary Figure 5B, and S6 is not mentioned in the manuscript. To make the setting more comprehensible and easy to read, the authors must homogenize it.
Response: Reference to the Figure S4 could be found on line 181 (see our response to the Reviewer’s #2 comment #3) and line 450 in the original submission. “Supplementary Figure 5B” has been changed to the “Supplementary Figure S5”. Supplementary Figure S6 has been added to the sections “2.3. Effects of 3D printed cells in fibrin hydrogel on neurite outgrowth” abd “4.10. Neurite outgrowth assay”.
Minor issues
- Figure, figure
Response: We have change “figure” to “Figure”.
- Extra spaces between words
Response: We have checked the text for extra spaces between words.
- Avoid using sentences like "This observation may be in some part due to the presence of MSCs."
Response: The sentence has been corrected.
- Do the authors refer to Figure 4A in line 522?
Response: Figure 5 (line 522 in the original submission) has been changed to Figure 4A.
- The authors have ended the discussion with the statement, "This biomaterial can then be used as a biosynthetic conduit to promote axonal regenration after peripheral nerve injury." While this statement is not entirely appropriate and relevant to the findings, it is advised to rewrite it as "exclude" to avoid misunderstandings. There is no conclusion provided.
Response: The last paragraph in the Discussion has been revised and moved to the new section ”5. Conclusion”.
Round 2
Reviewer 1 Report
All my issues have been addressed